# Everyday Memory in People with Down Syndrome

**DOI:** 10.3390/brainsci11050551

**Published:** 2021-04-27

**Authors:** Yingying Yang, Zachary M. Himmelberger, Trent Robinson, Megan Davis, Frances Conners, Edward Merrill

**Affiliations:** 1Department of Psychology, Montclair State University, Montclair, NJ 07043, USA; 2Division of Behavioral Sciences, Maryville College, Maryville, TN 37804, USA; zach.himmelberger@maryvillecollege.edu; 3Department of Psychology, University of Alabama, Tuscaloosa, AL 35487, USA; atrobinson2@crimson.ua.edu (T.R.); mjbenson@as.ua.edu (M.D.); fconners@ua.edu (F.C.); emerrill@ua.edu (E.M.)

**Keywords:** everyday memory, down syndrome, intellectual disabilities, verbal memory, list learning, ecologically valid

## Abstract

Although memory functions in people with Down Syndrome (DS) have been studied extensively, how well people with DS remember things about everyday life is not well understood. In the current study, 31 adolescents/young adults with DS and 26 with intellectual disabilities (ID) of mixed etiology (not DS) participated. They completed an everyday memory questionnaire about personal facts and recent events (e.g., school name, breakfast). They also completed a standard laboratory task of verbal long-term memory (LTM) where they recalled a list of unrelated words over trials. Results did not indicate impaired everyday memory, but impaired verbal LTM, in people with DS relative to people with mixed ID. Furthermore, the laboratory verbal LTM task predicted everyday memory for both groups after taking into account mental age equivalent. Our research showed both an independence and a connection between everyday memory and the standard laboratory memory task and has important research and clinical implications.

## 1. Introduction

Down syndrome (DS) is the most common genetic cause of intellectual disability and is caused by all or part of an extra 21st chromosome. Memory is an important and diverse cognitive ability that has received a great deal of research attention in people with DS (for a review, see [1]). The current study examined everyday memory in people with DS. Here we use the term everyday memory to refer to an ability to remember relevant facts and episodes of one’s daily activities. For instance, in daily life, we need to remember semantic and factual information, such as a loved one’s birthday or the name of important people in our daily lives (e.g., teacher or employer). It is also useful to remember episodic events such as what we had for breakfast or what gift we received last Christmas.

Before discussing everyday memory, we first need to discuss laboratory measures of verbal memory in people with DS. A large portion of that research has shown that people with DS exhibit deficiencies in verbal long-term memory (LTM) measured by standard laboratory tests such as the word list learning task (e.g., [2,3,4]). In the typical word list learning task, children are presented a list of unrelated words and asked to recall either immediately or after some delay. Using this general task, Carlesimo and colleagues [5] found that people with DS recalled fewer words than expected based on their mental age equivalent (MA, as measured by WISC-R or WAIS) in comparison to people with an intellectual disability (ID) of unknown etiology and typical developing (TD) children. This pattern of impaired verbal LTM has been found numerous times by different researchers (e.g., [2,3,4,5]). However, because the words are usually unrelated to each other in the word list learning task, there is no context for recall of the words.

Memory of meaningful and related materials can be assessed in the laboratory via the Rivermead Behavioral Memory Test (RBMT; [6]). It includes a subtest called “route recall” in which participants retrace a route in a room shown by the experimenter, and a task called “story recall” in which participants recall a short story they have just heard. Numminen and colleagues [7] found no difference between 15 participants with DS and participants with ID of mixed etiologies matched on non-verbal MA in RMBT. However, in addition to the small sample size, participants with DS in this study also had a mean chronological age (CA) of over 42 years old. Hence, they were at risk for and might have already experienced cognitive decline even in the absence of clinical diagnosis of dementia [8], thereby qualifying conclusions about memory in people with DS in general. There are two other reports of RMBT for persons with DS [6,9], but neither study included an MA-matched comparison group. Therefore, it was not possible to determine whether the performance on RMBT was impaired in people with DS relative to their MA. Although somewhat more experience-based than the word list learning task, the RMBT task still consists of new and novel materials (e.g., route, story) that participants have never encountered before. Therefore, the materials were still different from common everyday memories.

In the research presented here, we were primarily interested in everyday memory in a contextualized setting. Although everyday memory can rely heavily on verbal report, it is distinguished from verbal memory by its rich context and personal relevance [10,11]. Lately, there has been a resurgent interest in everyday memory in people with DS. Spanò and Edgin [12] validated a parent report of everyday memory in children with DS. In their questionnaire, parents reported their children’s behaviors in everyday life, such as remembering where they put their things, their beliefs about how well their children remember things in general and related complaints. They found that parent reports correlated significantly, although modestly, with children’s performance on the CANTAB PAL Test, a standard laboratory test of spatial associative memory. The authors also compared reports from parents of children with DS (mean children CA: 18 years old) with parents of MA-matched TD children (mean CA: 5). Everyday memory for children with DS was reported to be 3 standard deviations (SD) lower than that of TD children. Furthermore, there was a non-significant trend (*p* = 0.06) such that the age of the participants was positively associated with everyday memory.

In contrast to the parent reports of memory presented in Spanò and Edgin [12], Pennington et al. [2] used an 18-item “ecological questionnaire” to directly assess participants’ Everyday Memory. In this questionnaire, participants needed to answer questions such as parents’ names, waking and bed times, recent meals, birthday, and what they did in a previous testing session. Overall, participants with DS (mean CA: 15) performed at a similar level as MA-matched TD children (mean CA: 5). Interestingly, children with DS performed worse when recalling the activity they did in the previous testing session yet outperformed the TD children at recalling the examiner’s name. In the same study, Pennington et al. also found that people with DS performed worse in the standard laboratory LTM measures, including the word list learning task and the Morris water maze task. This research is notable because it suggests that people with DS may not have a general deficit in everyday memory, despite marked impairments in verbal and spatial LTM measured using standard laboratory tests.

## 2. Current Study

The goal of the present study was to examine everyday memory in people with DS relative to persons with mixed ID and to investigate possible relations between a laboratory measure of verbal memory and everyday memory. Our study contributes to the knowledge of memory in DS in two ways. First, rather than employing TD children as a comparison group, we recruited and tested participants with ID of mixed etiologies (other than DS) as a comparison group. This method is particularly helpful in determining whether the everyday memory profile exhibited by persons with DS is specific to their syndrome or represents a more general deficiency associated with ID. Unlike TD children, adolescents and adults with mixed ID are usually older and comparable in age relative to people with DS. They have more semantic and linguistic knowledge relative to MA matched 5–7-year-old TD children who have less life experience [13,14], and may therefore be at a disadvantage.

The second contribution of this study is that we also examined the possible relation between a laboratory measure of verbal memory (via word list learning) and everyday memory. Pennington et al. [2] did not explore this relation despite testing participants on both types of measures. Although Spanò and Edgin [12] found a significant correlation between everyday memory and a laboratory test of spatial associative memory, their measure of everyday memory was via parent report, and not a direct measure of performance of individuals with DS. In the present study, we examined how effective the laboratory test of verbal memory is at predicting everyday memory performance. After all, one ultimate goal for many laboratory tests/experiments is to inform everyday clinical practice and help improve daily functioning for people with DS. From a theoretical perspective, the correlation between the two measures may help us understand how memory of new materials may contribute to memory consolidation over an extended period of time as in everyday memory. It may, in turn, have clinical implications for training and intervention programs to improve memory performance in people with DS. In the current study, everyday memory was examined via a questionnaire where participants recalled a series of real-life events or facts such as teachers’ names, meals in the last 24 h, and bedtime. The laboratory test was a word list learning task, which mainly measured verbal LTM.

## 3. Method

### 3.1. Participants

Participants were 31 adolescents/young adults with DS and 26 with ID of mixed etiology who were recruited to participate in two larger studies. See Table 1 for participant details. Other data from these studies are reported elsewhere [15]. For the present project, we included all participants who completed both the Word List Learning task and the Everyday Memory tasks. The final groups consisted of 19 males and 12 females with DS, and 14 males and 12 females without DS. All participants were required to meet the following criteria, based on parent report: (1) adequate verbal skills for understanding instructions and for providing verbal responses, and (2) no accompanying diagnoses of Autism Spectrum Disorder as autism symptomology could confound results. Participants with DS were also required to have documented chromosomal analysis to indicate Trisomy 21. For the group of mixed ID, 16 participants had ID of unknown etiology, 3 participants had Fragile-X syndrome, 2 had ID of postnatal cause (malnutrition, Traumatic Brain Injury), 1 had XXYY syndrome, 1 had Rubinstein–Taybi syndrome, and 3 had Fetal Alcohol Syndrome.

### 3.2. Measures

#### 3.2.1. Nonverbal MA

Nonverbal MA was based on either the Leiter-R brief form (19 DS and 20 ID participants) or the KBIT-2 Matrices subtest (12 DS and 6 ID participants). This was due to using different measures to identify MA in the two larger studies. Four subtests of Leiter-R were administered: Figure Ground, Form Completion, Sequential Order, and Repeated Patterns. These subtests measure visuospatial and inductive reasoning skills typically classified as fluid intelligence. The KBIT-2 Matrices subtest consists of a 2 × 2 or 3 × 3 grid of pictures with one element missing. Participants are asked to choose which one of five pictures best completes the grid. The KBIT-2 correlates well with the Leiter-R (*r* = 0.62) in children with special needs [16]. Results of analyses did not change when including all participants relative to including only participants whose MA was based on the Leiter-R.

#### 3.2.2. Word List Learning

Our task was patterned after the NEPSY List Learning Test as adapted by Pennington et al. [2]. Participants were presented a list of 15 one-syllable words by the experimenter (e.g., ball, pen, shoe, box, shovel, doll) at a rate of one word per second. After the presentation of the entire list, participants were asked to recall as many words as possible. The list, followed by an immediate recall, was repeated a total of five times. Hence, scores could be obtained for each recall attempt (max = 15) and the total number of words recalled across attempts (max = 75). While the first attempt may be deemed short-term memory (STM), the total number of words over 5 trials is deemed LTM. The word-learning task was highly reliable, reliability coefficient *r* = 0.91 (split-half and alpha methods) [17].

#### 3.3.3. Everyday Memory

Ten questions of everyday memory, including own birthday, teachers’ name, phone number, street name, breakfast, bedtime, afterschool activity, type of pets owned, number of siblings, and school name, were asked of the participant and one parent. The participant and one parent answered the questions separately. Participants’ responses were judged correct if they provided the same response as the parent. The total number of correct responses was the dependent measure. The reliability is good, Cronbach’s alpha = 0.736.

## 4. Results

### 4.1. Preliminary Analyses

See Table 1 for the descriptive statistics. CA did not significantly differ between the two groups. However, MA was significantly higher in the ID group than that in the DS group, *F*(1,55) = 16.947, *p* < 0.001. Thus, MA was treated as a covariate in analyses where appropriate. For two dependent variables, skewness and kurtosis were also listed. They were all between +/−1, indicating normality. Moreover, if we define outliers as 3 SD outside the mean, there were no outliers in either verbal LTM or everyday memory for either the DS or the mixed ID groups. See Table 2 for the correlations between variables. MA was significantly correlated with everyday memory for each group separately and for the entire sample. MA was correlated with verbal LTM for the participants with ID, but not for participants with DS. No correlations with CA were significant. Hence, CA was not considered in the following trajectory and regression analyses. Finally, verbal LTM and everyday memory were significantly correlated for each group separately and for the entire sample.

### 4.2. Everyday Memory

#### 4.2.1. Total Score

We converted raw scores to z scores using the overall mean and standard deviation of the full sample. We then regressed everyday memory on MA for the DS and mixed ID participants separately. These are shown in Figure 1. MA accounted for 15% of variance in everyday LTM (*p* = 0.053) for persons with mixed ID and 12% of variance (*p* = 0.056) for person with DS.

Next, we used cross-sectional developmental trajectory analyses ([18]; for example, see [19,20]) to examine everyday memory as a function of MA. This approach is effective at describing how performance varies with developmental levels across different groups. We first rescaled MA by subtracting 46 months (the minimum MA across both groups) from the MA of each participant to obtain the rescaled MA. A univariate ANCOVA was conducted where rescaled MA was entered as the covariate and the interaction term between rescaled MA and group was also included. A significant interaction between rescaled MA and group would indicate different trajectory slopes between different groups. A significant main effect of group would indicate different trajectory intercepts between two groups. The model explained a significant proportion of variance, *F*(3,53) = 4.63, *p* = 0.006, η^2^_p_ = 0.208. The main effect of rescaled MA was significant, *F*(1,53) = 7.73, *p* = 0.008, η^2^_p_ = 0.127, as expected. However, the main effect of group was not significant, *F*(1,53) = 1.32, *p* = 0.214, indicating no evidence that the intercepts/onsets of developmental trajectories differed between the two groups. The interaction between rescaled MA and group was not significant, *F*(1,53) =1.59, *p* = 0.214, η^2^_p_ = 0.029, (or Cohen’s d = 0.34), indicating no evidence that the slopes/gradients of developmental trajectories differed between the two groups.

#### 4.2.2. Individual Items

First, we explored whether some questions were answered correctly more often than others for each group. For ease of interpreting any similarities and differences between groups, we matched a subsample of participants with DS and with mixed ID as nearly as possible on MA. We included an equal number of participants per group that fell between an MA of 48 months (the lowest MA of the ID group) and 90 months (the highest MA of the DS group). Participants with DS were individually paired with the closest match possible in the ID group. This resulted in 18 closely matched pairs with a mean MA of 64.0 months for the DS group and 66.7 months for the ID group. See Table 3 for accuracies for each question in the matched groups (first two columns). The average accuracy for the top four questions was 81% for the DS group and 90% for the ID group. The average accuracy for the bottom four questions was 60% for the DS group and 61% for the ID group.

Second, we focused on the entire sample in terms of accuracy and rank order of each question. As seen in Table 3, The top four questions were the same for the matched groups and the entire samples of both participants with DS and with mixed ID. The bottom four questions are the same for the matched sample and the entire sample of participants with ID. For participants with DS, the bottom four questions were all ranked after the top four questions. However, there was one deviation among the bottom four questions. The question regarding phone number was ranked the lowest when examining the entire sample of DS participants but not when only examining the subsample of DS participants. This may be partly due to the lower MA of those participants who could not be matched. They may have had particular difficulty with rote memory, and hence remembering their phone number.

Third, we analyzed individual performance to see whether each participant showed a pattern consistent with the group. We summed the number of Top four questions answered correctly and the number of Bottom four questions answered correctly by each individual. We then classified each participant as TOP (answered more of the Top four questions correctly than Bottom four questions), BOTTOM (answered more of the Bottom four questions correctly), or NEUTRAL (answered the same number correctly for the Top and Bottom four questions). These data are presented in Table 4. Groups were compared using a Chi-Square Test of Independence. The analysis indicated that the proportion of participants in each classification were essentially the same for both groups, χ^2^(2) = 0.32, *p* = 0.85. After combining groups, a second Chi-Square test revealed a significant difference among the three categories, χ^2^(2) = 24.0, *p* < 0.001, with far more participants identified in the TOP (35) than in the BOTTOM (5) category or NEUTRAL category (17). Therefore, most participants showed a response pattern consistent with the group.

Fourth, we concentrated on the more episodic questions. There were three questions we classified as episodic: “What did you have for breakfast today?”, “When did you go to bed last night?”, and “What did you do after school/work on Friday?” We were interested in whether there was a group of participants with DS or ID that systematically passed or failed all of the more episodic questions. To evaluate this question, we summed the accuracy for the three episodic questions for each individual (range = 0 for all incorrect to 3 for all correct). See Table 5 for the frequency distribution. There are participants at each possible score. Chi-square goodness of fit test suggested no difference in the number of participants across the four possible scores, χ^2^(3) = 3.14, *p* = 0.37. Therefore, there is no evidence that episodic memory is “all or nothing” for participants with either DS or ID.

### 4.3. Word List Learning Task

#### 4.3.1. Total Score

We converted raw scores to z scores using the overall mean and standard deviation of the full sample. We regressed verbal LTM on MA for the DS and mixed ID participants separately. These are shown in Figure 2. MA accounted for 21% of the variance in verbal memory (*p* = 0.018) for persons with mixed ID. However, MA made a non-significant 1% contribution in verbal LTM for persons with DS. One might suspect the role of outliers in the lack of relation between MA and verbal LTM. We examined Cook’s D, a measure of how much a single data point exerts disproportionate influence on the regression. A Cook’s D of over 1.00 suggests that a given data point is a potential outlier [18]. Results suggested that Cook’s Ds were less than 0.35 for all participants in either group. Hence, the lack of relationship should not be due to the influence of outliers.

Because linearity cannot be established for the DS group, it is not appropriate to directly compare two groups using developmental trajectory analyses. We used the rotation method by Thomas et al. [18] to explore whether the trajectory slope was zero or whether there was simply no systematic relation between MA and verbal LTM for people with DS. If cognitive factors other than MA predicted verbal LTM and hence there was no systematic relation between MA and verbal LTM, then rotating the trajectory by 45 degrees should result in similarly small R^2^. However, if participants had reached the best level they could achieve and that the trajectory slope was “truly” zero, then rotating it by 45 degrees should result in relatively larger R^2^. Indeed, after rotating the trajectory by 45 degrees, we found that R^2^ rose to 53.9%, *p* < 0.001. Therefore, people with DS generally performed at a low level in verbal LTM regardless of their measured MA.

To examine the group difference in average scores, we conducted an ANCOVA with MA as a covariate. The main effect of group was significant, *F*(1,54) = 5.50, *p* = 0.023, η^2^_p_ = 0.092, (or Cohen’s d=.64, an intermediate effect) with better performance in the mixed ID group (Adjusted Mean = 0.341, covariate MA = 66.88) than the DS group (Adjusted Mean = −0.286, covariate MA = 66.88). The main effect of MA was also significant, *F*(1,54) = 5.01, *p* = 0.029, η^2^_p_ = 0.085, as expected.

#### 4.3.2. Individual Attempts

In an attempt to better understand the source of the group difference in verbal LTM, we evaluated whether group differences in recall varied as a function of recall attempts using a 2 (Group: DS vs. mixed ID) × 5 (Recall Attempt) ANCOVA with recall attempt treated as a within-subjects factor and MA as a covariate. The analysis revealed a significant main effect of group, *F*(1,54) = 5.46, *p* = 0.023, η^2^_p_ = 0.092, with participants with ID recalling more words on average (Adjusted M = 5.86, covariate MA = 66.88) than participants with DS (Adjusted M = 4.59, covariate MA = 66.88). The main effect of recall attempt did not reach significance, but exhibited a tendency to do so, *F*(3.12,168.85) = 2.30 (Greenhouse–Geisser), *p* = 0.077, η^2^_p_ = 0.041, indicating participants generally recalled more during latter attempts.

In addition, the interaction between recall attempt and group was significant *F*(3.13,168.85) = 2.91, *p* = 0.034, η^2^_p_ = 0.051. Tests of simple effects indicated that for people with ID, recall significantly improved from attempt 1 to attempt 2, and from attempt 2 to attempt 3, *p*s < 0.01. However, the improvement from attempt 3 to attempt 4 and from attempt 4 to attempt 5 was not significant, *p*s > 0.09. For people with DS, recall significantly improved from attempt 1 to attempt 2, *p* < 0.01. However, attempts 2, 3, 4, and 5 did not differ from each other, *p*s > 0.18. Tests of simple effects also indicated that participants with ID recalled more items than participants with DS on Recall Attempts 1, 3, 4, and 5 (all significant *ps* < 0.05). As seen in Table 6, participants with DS performed very poorly on the first recall attempt, caught up slightly on the second attempt, but lost ground to the participants without DS on all subsequent attempts. Hence, the participants with DS not only recalled fewer items initially, but they did not benefit as much from future exposure to the items.

### 4.4. Relationships between Everyday Memory and Verbal Memory

First, we used scatterplots to graph everyday memory as a function of verbal LTM for the DS and ID groups separately. See Figure 3. For both groups, increased verbal LTM was associated with better everyday memory.

Next, we used hierarchical regression to identify significant predictors of everyday memory for the full sample. We entered MA first, verbal LTM second, and group status (DS vs. ID) last. We were interested in whether verbal LTM could account for variance after considering the effect of MA. We were also interested in whether the two groups showed differences in everyday memory after accounting for MA and verbal LTM. For both everyday memory and verbal LTM, z scores were used. This way, the intercept (constant) can also be interpreted. The model summaries are presented in Table 7. Verbal LTM predicted additional variance in everyday memory above and beyond those already accounted for by MA. Furthermore, after MA and verbal LTM were accounted for, group status did not differentiate everyday memory performance.

Then, we explored the relationship between verbal LTM and episodic memory. We regressed the total score of episodic memory (highest possible score: 3) on MA, verbal LTM, and group status using the same hierarchical regression method mentioned above. Although MA was significant in the first step, β = 0.281, *p* = 0.034, adjusted R^2^ = 0.08, it was no longer significant in the second step when verbal LTM was included, β = 0.129, *p* = 0.362. Verbal LTM was significant in the second step, β = 0.329, *p* = 0.024, R^2^ change = 0.08. Group status was not significant in the 3rd step, β = 0.164, *p* = 0.287. Hence, verbal LTM still predicted episodic memory score. Next, we focused on the other more semantic questions of the everyday memory questionnaire. We regressed the total score of semantic memory (highest possible: 7) on MA, verbal LTM, and group status using the same hierarchical regression method mentioned above. MA was significant in the first step, β = 0.417, *p* = 0.001, adjusted R^2^ = 0.16, and marginally significant in the second step when verbal LTM was included, β = 0.259, *p* = 0.054. Verbal LTM was significant in the second step, β = 0.340, *p* = 0.013, R^2^ change = 0.09. Group status was not significant in the 3rd step, β = −0.075, *p* = 0.605. Hence, verbal LTM still predicted semantic memory score.

Finally, we considered the role of verbal STM indicated by the first recall attempt in the word list learning task on everyday memory performance (total score). We entered MA first and verbal STM second in the regression model of everyday memory. However, including STM in the second step only increased the R^2^ by 4.4%. The regression coefficient for STM was B = 0.141 (SE = 0.08), β = 0.237, *p* = 0.084. Hence, verbal STM played a lesser role in everyday memory relative to verbal LTM.

## 5. Discussion

The current study examined everyday memory in people with DS relative to people with mixed ID, as well as the relation between everyday memory and a laboratory task of verbal LTM for the two groups. Our results provided no evidence that people with DS were impaired in everyday memory relative to people with mixed ID. Both groups showed relatively better performance in semantic everyday memory questions than in episodic everyday memory questions. Furthermore, people with DS were impaired in the laboratory task of verbal LTM relative to people with mixed ID. Everyday memory was significantly associated with MA for both people with DS and with mixed ID. Verbal LTM correlated with MA for people with mixed ID, but not for people with DS. Finally, the laboratory task of verbal LTM predicted the total everyday memory above and beyond the effects of MA for people with DS and mixed ID. Verbal LTM also predicted performance on both episodic and semantic everyday memory questions for people with DS and mixed ID.

### 5.1. Comparing People with DS and ID on Everyday Memory

People with DS and with mixed ID exhibited a general similarity in everyday memory performance. The two groups showed no difference in the developmental trajectories of everyday memory as a function of MA. Furthermore, being in the DS or mixed ID groups did not predict everyday memory above and beyond the effects of MA and verbal LTM. Therefore, everyday memory may not be an area of impairment for people with DS relative to a mixed ID comparison group.

In reviewing the individual questions of the everyday memory questionnaire, we can see both groups’ performance depended on the type of information that was asked. High accuracy questions included information that was likely repeated many times over a very long period, including school/work name, teacher/boss name, pets owned, and their birthday– perhaps reflecting facts and a semantic memory quality. A review by Conners and colleagues [21] concluded that people with DS were impaired in some aspects of semantic memory, such as gist recall of stories, but not impaired in other aspects of semantic memory such as rapid retrieval of semantic knowledge. Our study showed that people with DS and mixed ID showed relatively high accuracy (over 80%) in the semantic memory of personal facts.

Three of the four low accuracy questions included information that was more episodic. They were about personally experienced events, having occurred at a single time in the past at a particular location (i.e., recent breakfast, recent bedtime, Friday after school/work). As for semantic memory, research is mixed regarding whether people with DS show impairment in episodic memory. Some research of episodic memory using laboratory tasks (e.g., “Picture Sequence Memory”, “People Test”, “Name Test”) found no difference between people with DS and MA matched controls, including people with ID of unknown etiology (e.g., [22,23]). However, other research also using laboratory tasks (“Prose Recall” (story recall), “Rey’s Figure Form reproduction”) found worse episodic memory in people with DS relative to people with ID of unknown etiology (e.g., [5]). Our study found that both people with DS and people with mixed ID were relatively less accurate at recalling episodic memory of everyday life events (around 60%). The individual analyses revealed that most participants showed the pattern of worse episodic than semantic everyday memory, consistent with the group performance. Furthermore, this pattern was found not only in the MA-matched subsamples, but also in the entire samples of participants with DS and mixed ID.

Adult lesion studies, animal model research, and neuroimaging studies have found that both the hippocampus and the frontal lobes contribute to episodic memory and semantic memory [24,25,26,27,28,29]. Meanwhile, both hippocampal and frontal lobe abnormality have been well documented in DS [30,31,32,33,34,35]. The specific cognitive profile of LTM, including episodic and semantic memory, exhibited by people with DS presumably may result from their anomalous brain development [36]. However, it is hard to map the LTM probed in our study to specific brain regions precisely. People with DS may still show impairments in LTM in certain domains, especially when using laboratory tasks. However, our study using everyday memory suggests that at least people with DS may perform similarly to people with mixed ID on recollecting personal facts and episodic events.

It is peculiar that the number of siblings was also associated with relatively low accuracy for both participants with DS and mixed ID. There are some possible reasons, such as confusion about half-siblings, trouble with counting, counting imaginary friends, and of course, difficulty remembering. It is also possible that participants know the number of brothers and sisters separately, but not the sum. Unfortunately, the everyday memory questionnaire did not contain follow-up questions, and future studies may further investigate this issue.

### 5.2. Laboratory Task Predicted Everyday Memory

One of the important goals of this manuscript is to see how effective a widely used laboratory task of verbal LTM (word list learning) is at predicting everyday memory, a contextualized measure with more ecological validity. Traditional neuropsychological tests have been criticized for evaluating abstract constructs without any reference to real-life performance or behavior [37,38], and for being mainly construct- and theory-driven. There has been a growing interest in assessing the more ecologically valid function of memory. This calls for a more function-driven approach focused on elucidating the specificities and complexity of memory use in daily life [39].

Our study indeed showed both a connection and a clear distinction between everyday memory and the laboratory task of verbal LTM. On the one hand, the laboratory task of verbal LTM significantly predicted everyday memory, even after taking MA into account, indicating the predictive power was robust. Furthermore, verbal LTM also significantly predicted performance on both episodic and semantic memory questions, after accounting for MA. Because everyday memory was assessed via verbal recall, verbal memory search and retrieval processes may have been similar for both memory tasks. More importantly, both share the common mechanisms of encoding, storage, retrieval, and consolidation of LTM. On the other hand, the word list learning task, which is frequently used to indicate participants’ verbal LTM ability, only accounted for a meager 12% of the total variance of how participants remember everyday events and facts, indicating the predictive power is also limited at the same time (these two sides are analogous to our interpretation of significance levels vs. effect sizes). Moreover, verbal LTM predicted an additional 8% of the variance in episodic memory questions and an additional 9% of the variance in semantic memory questions, after accounting for MA. It hence suggested a rather similar role of verbal LTM in both types of everyday memory.

As suggested by Tulving [40], traditional Ebbinghaus-inspired study/test laboratory experiments, as in our word list learning task, are almost invariably concerned with “what” has been remembered. However, the everyday memory questions such as what they had for breakfast and afterschool activities concern not only “what”, but also “where” and “when”. The everyday memory questions involve the personal experience, events, and knowledge of the participants invoking the concept of self, whereas traditional laboratory tasks do not. Furthermore, everyday memories of personal facts are formed over long-term repeated exposures. In contrast, verbal LTM formed through list learning involves repeated exposures over a relatively short time span (e.g., minutes). Memory is not a uniform construct but rather multifaceted. Tulving [40] described a patient K.C. who suffered extensive brain lesions, including medial temporal lobes and consequent severe amnesia. K.C. could learn new factual information such as 3-word sentences and word definitions and retain it over weeks and months, but could not recollect any visits to the laboratory (i.e., episodic events). Our study shows a seemingly opposite pattern for people with DS in that they are able to remember personal events and knowledge, but have more trouble with remembering word lists. One important clinical implication is that how people with DS do with everyday memory may depend on the type of information (e.g., episodic vs. semantic) they are trying to remember. Additionally, given sufficient time to learn and/or given personally relevant events/experience/facts, people with DS may be able to exhibit an everyday memory as good as others of a similar cognitive level [2].

The impairment in laboratory verbal LTM performance in people with DS was shown in at least three forms. First, comparing people with DS and ID showed that people with DS performed worse relative to people with mixed ID. Second, MA was not correlated with verbal LTM in persons with DS. Hence, in addition to affirming a verbal LTM deficit for persons with DS as reported by Carlesimo et al. [5], Nichols et al. [4], and Pennington et al. [2], our results indicate that the deficiency exists across a range of nonverbal MA. Third, the number of words recalled over trials increased more slowly for people with DS than for people with mixed ID. Taken together, the slower improvement observed for the participants with DS over trials and with increases in MA may reflect their impairments in memory rehearsal [41], phonological loop [42], memory consolidation and retrieval [43], and other executive function processes [44] such as monitoring previously recalled items and selectively attending to missed items.

## 6. Conclusions

Our study of everyday memory in people with DS contributes to the existing literature in two ways. First, we found no evidence that people with DS performed differently in everyday memory than people with mixed ID, indicating that everyday memory may be a relatively preserved area of memory function for people with DS. Second, we found that the laboratory task of verbal LTM predicted everyday memory above and beyond the effects of MA. This showed both the independence and connection between the laboratory task of verbal LTM and the more ecologically valid task of everyday memory. Memory is one of the quintessential human abilities. Recognizing that memory is multifaceted, flexible, relational, and contextualized [25], it is essential to fully capture the breadth and depth of human memory. Studying everyday memory and its relation to laboratory tasks of memory may help us better understand memory in a contextualized lens, the cognitive dependencies between different forms of memory, and how memory may support cognition in general.

## Figures and Tables

**Figure 1 brainsci-11-00551-f001:**
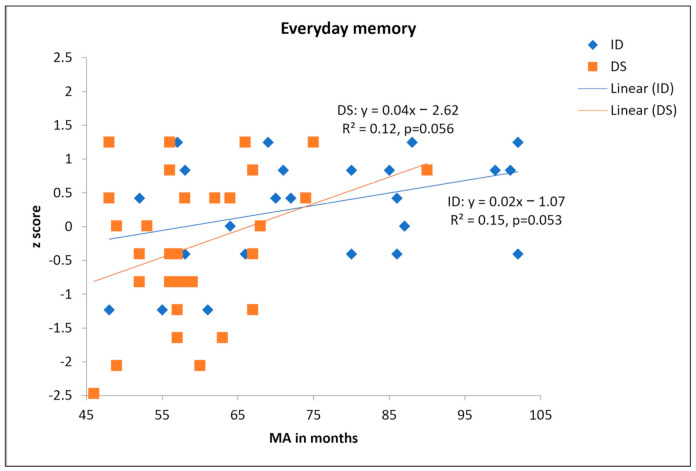
Scatterplot of z scores in everyday memory as a function of MA.

**Figure 2 brainsci-11-00551-f002:**
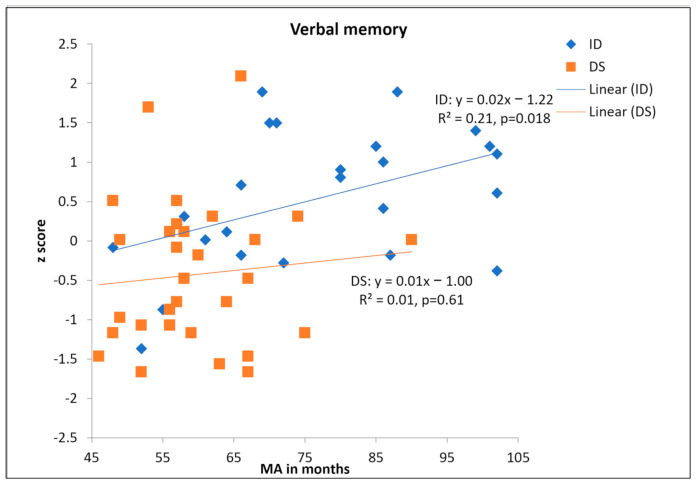
Scatterplot of z scores in verbal memory as a function of MA.

**Figure 3 brainsci-11-00551-f003:**
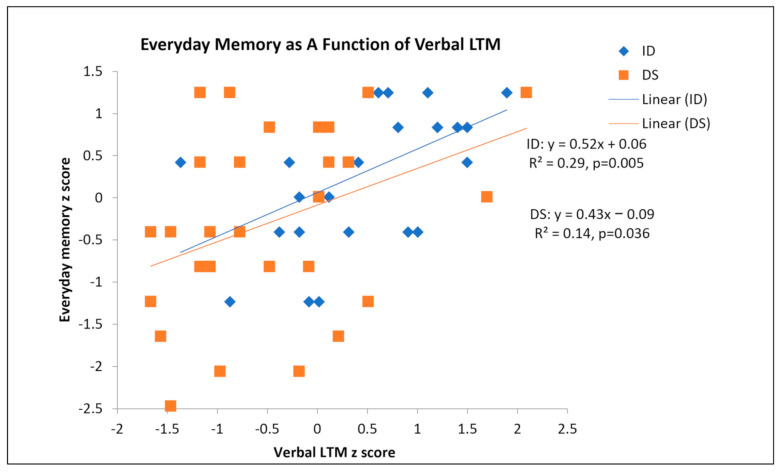
Scatterplot of everyday memory as a function of verbal LTM.

**Table 1 brainsci-11-00551-t001:** Descriptive statistics.

	Mean	SD	Range	Skewness	Kurtosis
CA (months, with years in parentheses)					
DS	221.50 (18.46)	44.00 (3.67)	126–309 (10.5–25.75)	-	-
ID	215.00 (17.92)	54.81 (4.57)	150–310 (12.5–25.83)	-	-
MA (months, with years in parentheses)					
DS	58.84 (4.90)	9.32 (.78)	46–90 (3.83–7.50)	-	-
ID	75.58 (6.30)	17.42 (1.45)	48–102 (4–8.5)	-	-
Everyday memory (total score)					
DS	6.32	2.57	1–10	−0.30	−0.78
ID	7.77	1.99	4–10	−0.62	−0.72
Verbal LTM (total score)					
DS	21.52	9.35	9–47	0.86	0.77
ID	31.00	8.59	12–45	−0.25	−0.53

**Table 2 brainsci-11-00551-t002:** Correlations between MA, CA, Everyday Memory, and Verbal Memory.

	Group	CA	Everyday Memory (Total Score)	Verbal LTM (Total Score)
	All	−0.03	0.43 **	0.46 **
MA	DS	0.03	0.35 ^t^	0.10
	ID	−0.01	0.38 ^t^	0.46 *
	All		0.15	−0.15
CA	DS		0.16	−0.16
	ID		0.21	−0.1
Everyday	All			0.51 **
memory	DS			0.38 *
	ID			0.54 **

Note: ^t^
*p* < 0.06; * *p* < 0.05; ** *p* < 0.01.

**Table 3 brainsci-11-00551-t003:** Accuracy (rank order in parentheses) on Everyday Memory questions for the matched subsamples and the whole samples of mixed ID and DS participants.

Item	Matched DS (*n* = 18)	Matched ID (*n* = 18)	Total DS (*n* = 31)	Total ID (*n* = 26)
*Where do you go to school/work?*	*0.78(2)*	*0.94(1)*	*0.77(2)*	*0.96(1)*
*What is the name of your teacher/boss?*	*0.78(2)*	*0.89(3)*	*0.74(3)*	*0.88(3)*
*What kinds of pets do you have?*	*0.78(2)*	*0.94(1)*	*0.71(4)*	*0.96(1)*
*When is your birthday?*	*0.89(1)*	*0.83(3)*	*0.84(1)*	*0.88(3)*
What is your phone number?	0.61(7)	0.72(5)	0.42(10)	0.77(5)
What is the name of your home street?	0.72(5)	0.67(6 *)	0.58(6)	0.73(6)
**What did you have for breakfast today?**	**0.67(6)**	**0.61(9)**	**0.65(5)**	**0.65(8)**
**How many brothers and sisters do you have?**	**0.61(8)**	**0.67(6 *)**	**0.58(6)**	**0.69(7)**
**When did you go to bed last night?**	**0.56(9)**	**0.50(10)**	**0.45(9)**	**0.62(9)**
**What did you do after school/work on Friday?**	**0.56(9)**	**0.67(6)**	**0.55(8)**	**0.62(9)**
**Total accuracy**	**0.70**	**0.74**	**0.63**	**0.78**

Note: The top four questions (italicized) represent the four highest accuracy questions for both groups. The bottom four questions (bold printed) represent four of the lowest ranked questions (rank 6 or below) for both groups. The two middle questions fell on opposite sides of the midpoint of accuracy for the two groups. * Tie had no bearing on analysis or interpretation.

**Table 4 brainsci-11-00551-t004:** Down syndrome and mixed ID participants classified by types of questions most often answered correctly.

Group	TOP	BOTTOM	NEUTRAL
DS	18	3	10
ID	17	2	7
Total	35	5	17

**Table 5 brainsci-11-00551-t005:** Number of Participants in Each Possible Episodic Memory Score (highest: 3).

	Score: 0	Score: 1	Score: 2	Score: 3
DS	6	7	10	8
ID	3	7	6	10
Total	9	14	16	18

**Table 6 brainsci-11-00551-t006:** Word List Learning Performance (mean total correct, with SD in parentheses) on Individual Attempts.

Group	Attempt 1	Attempt 2	Attempt 3	Attempt 4	Attempt 5
DS	2.52(1.52)	4.13(2.13)	4.48(2.16)	5.19(2.44)	5.19(2.91)
ID	4(1.5)	5.27(1.89)	6.35(2.08)	7.15(2.31)	8.23(2.42)

**Table 7 brainsci-11-00551-t007:** Regression Results where everyday memory is the DV.

Steps	Predictor	Unstandardized Coefficients		Standardized Coefficients	Sig.	F	Adjusted R^2^	∆R^2^
		B	SE	β				
1	-	-	-	-	-	12.41 ***	0.169	-
	(Constant)	−1.846	0.538	-	0.001			
	MA	0.028	0.008	0.429	0.001			
2	-	-	-	-	-	11.89 ***	0.280	0.122
	(Constant)	−1.060	0.562	-	0.065			
	MA	0.016	0.008	0.247	0.059			
	Verbal LTM	0.393	0.128	0.394	0.003			
3	-	-	-	-	-	7.79 ***	0.267	0.000
	(Constant)	−1.110	0.684	-	0.111			
	MA	0.016	0.009	0.254	0.076			
	Verbal LTM	0.398	0.135	0.399	0.005			
	Group Status	0.036	0.279	0.018	0.897			

Note: group status: 1: DS, 0: mixed ID, *** *p* ≤ 0.001.

## Data Availability

Data have been deposited in Montclair State University Digital Commons. https://digitalcommons.montclair.edu/psychology-facpubs/593/ (accessed on 26 April 2021).

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
