# Peer review of "Everyday Memory in People with Down Syndrome"

_brainsci, 2021, doi:10.3390/brainsci11050551_

Round 1

Reviewer 1 Report

  1. Interesting work, technically adequate.
  2. It adds to the existing literature on memory assessment suggesting that at least with people with Down syndrome the psychometric verbal LTM tasks lack in ecological validity. However, the work confirms that these tasks predict everyday memory scores after taking mental age into account. This simply means that who has the ability to do more can also do less.
  3. The authors should supply at least some minimal information on the etiologies of the mixed ID group used as a control. Using heterogeneous groups in comparative research with people with Down syndrome (or other etiological categories) should be discouraged . This type of matching increases the variance error in the control group and prevents identifying possible neurogenetic paths in the interpretation of the data.

Reviewer 2 Report

This is a concise, well-written manuscript that presents the analysis of data gathered from a previously conducted larger study. The authors show that (1) individuals with DS and ID performed similarly on an assessment of everyday memory (10 questions assessing mostly semantic knowledge about their life); (2) individuals with DS performed less well on a verbal LTM task (a 15-word task); and finally that (3) performance on the verbal LTM task correlated positively with everyday memory in these two populations, although with relatively small predictive value.

Although the data is clean, given the relatively little data and small effect sizes with respect to the correlation between the verbal LTM task and the everyday memory task (R2=.122), the overall importance of this finding to the general field is relatively limited. In contrast, whereas the data on everyday memory in both DS and ID of mixed etiology is novel and more reliable, the authors do not describe this data in sufficient detail. However, I feel that there is potential to increase the importance of the study by including more descriptions of the analyses of individual participants, as I describe in detail, and that these analyses are critical.

The author’s hierarchical regression analysis shows that verbal LTM could predict additional, albeit minimal, variance in everyday memory. I think that it is critical to show this data in a scatterplot (similar to Figures 1 and 2) so that the reader can get a better idea of how this data look.

The authors should better describe the data from Table 3 with respect to both the verbal LTM results and the overall everyday memory scores. Was there a group of individuals with DS or ID that systematically passed or failed all of the more episodic questions, and if so, how did they score on the verbal LTM task? For the 18 DS individuals in Table 3, the data suggest that between 56% and 67% correctly answered the episodic everyday memory questions. Were these more or less the same individuals? How did these specific individuals perform on the verbal memory task? Verbal memory tasks such as the one used in this study (and, for example, Rey’s Verbal Auditory Verbal Learning Test) are often considered to be source memory tests and have been shown to be sensitive to hippocampal dysfunction. Thus, describing how performance on these specific questions correlated with the verbal memory task in both groups of participants might be very informative. Moreover, these analyses should be done for all of the participants, not just in those matched for mean MA between the two groups (i.e., the 36 participants presented in Table 3). Given that this is relatively small paper with limited data, the data should be presented in as much detail as possible, especially the data concerning everyday memory, since this is the subject of the paper. Indeed, as the authors suggest, understanding whether the DS and ID individuals with high verbal memory capacity are the same as those with high everyday memory capacity, and in particular the same as those with greater episodic recall, will have important clinical implications for training and intervention programs to improve memory performance in these individuals, since these programs must take into account the capacities of each individual in order to adapt specific training to each person.

Other minor corrections that should be taken into account.

Abstract: line 14: change “personal and recent events” to “personal facts and recent events” since the name of the school, for example, is not an event.

Line 126: Please add: “See Table 1 for participant details”

Line 155-156: “The word learning task was highly reliable, r=.91”….what does this mean? Reliable as what? As compared to what?

Line 158 and elsewhere: Replace “child” with “participant” or “individual”

Table 1: Descriptive stats – use age in years to describe participants ages, not in months. Or, if you prefer to keep both, then please specify the age, and that it is in months, and then in parenthesis put the age in years, e.g., Mean: 221.50 mos. (18.5 yrs), etc.

Line 193-207: Cross-sectional developmental trajectory analyses. The authors state that they rescaled MA (mental age) by subtracting 46 (minimum score across both groups). Minimum score of what? Does this refer to the minimum age of all participants of both groups (46 months of age)? If so, this should be clarified. Age in months is not a score. If this is not the minimum months in age, then please explain what it is. Should be explained as: “We first rescaled MA by subtracting 46 months (the minimum MA across both groups) from the MA of each participant.”

Line 205-206: Please change “developmental trajectories differed in two groups” to “developmental trajectories differed between the two groups”. Same for line 208.

Line 243-244: “We used the rotation methods” should be either “We used the rotation method” or “We used rotation methods”.

Line 253: “we conduct” should be “we conducted”.

Line 256: Are the covariate MAs supposed to be the same for both the ID and DS groups (66.88)?

Line 264: Should read “recalling more words on average”

Line 266: “trending” should be “but exhibited a tendency to do so”.

Line 286: “in whether two groups” should read “in whether the two groups”.

Line 310: “Furthermore” should be replaced by either “Nevertheless” or “Finally”.

Line 344: “has been well documented” should be “have been well documented”.

Line 348: “areas” should be replaced by “domains”.

Line 349: “suggested” should be replaced by “suggests”.

Line 350-355: Number of siblings. How was this question asked to the participants, specifically? Is it the case that the participants were asked “How many brother and sisters do you have?”, when they might know the number of brothers and sisters separately, but not the sum. Or, were they asked, “How many brothers do you have?” and separately, “How many sisters do you have?”? Please provide details as to the specific manner in which this question was asked.

Line 359: The authors write that one of the goals of the manuscript is to see how a laboratory task of verbal LTM is at “predicting everyday memory, a contextualized measure with high ecological validity.” Whereas it is obviously hoped that such questionnaires have high ecological validity, what is the scientific evidence for supporting the idea that such questions have high ecological validity? Please cite references or rephrase this sentence to say that this is a “contextualized measure with more ecological validity.”

Line 362: “and they are mainly” should be replaced by “and for being mainly”

Line 354-357: The last sentence of this paragraph would read better as: “This calls for a more function-driven approach focused on elucidating the specificities and complexity of memory use in daily life.”

Line 410: “contributed” should be replaced by “contributes”.

Round 2

Reviewer 2 Report

I thank the authors for taking all of my comments into consideration during their revision. I believe that the paper now presents substantially more original data that can be used by other researchers to interpret their own findings or design new experiments that extend these findings.

Unfortunately, however, I have a few minor concerns regarding the revisions that must be addressed before the paper should be accepted for publication.

In Table 1, the word “parenthesis” should be “parentheses” (2X).

In Table 4, the title of the table should either read “…participants classified by type of question most often answered correctly” or “…participants classified by types of questions most often answered correctly”. All highlighted words need correcting.

In Figure 2, both regression lines are labelled as belonging to DS individuals. I believe one of these lines should be labelled as ID, no?

In Figure 3, please revise the figure using diamond-shaped markers for the DS group and square markers for the ID groups, as in Figures 1 and 2, and use the same types of regression lines as in Figures 1 and 2.

Also, please capitalize the word “Everyday” on the Y-axis.

Line 356: Please change the word “items” by “score”.

The authors did not incorporate any of their new analyses in the discussion. If I understand correctly, the authors’ new analyses show an equally strong predictive correlation between verbal LTM and Episodic memory score (lines 356-357), as between Everyday memory and verbal LTM. Please include a summary of the new data in the first paragraph of the Discussion, and include a brief discussion of this data where appropriate throughout the text of the discussion. Please answer specifically whether the data can be interpreted as showing that 3 episodic memory questions are as predictive of verbal LTM (or vice versa) as 7 more semantic-like Everyday memory questions.

Author Response

"Please see the attachment
